# XIRL:
# Cross-embodiment Inverse Reinforcement Learning

**Kevin Zakka**[1,3][*]**, Andy Zeng**[2]**, Pete Florence**[2]**, Jonathan Tompson**[2]**,**
**Jeannette Bohg**[1]**, and Debidatta Dwibedi**[2]
[1]Stanford University, [2]Robotics at Google, [3]UC Berkeley

**Abstract:** We investigate the visual *cross-embodiment* imitation setting, in which agents learn policies from videos of other agents (such as humans) demonstrating the same task, but with stark differences in their *embodiments* – shape, actions, end-effector dynamics, etc. In this work, we demonstrate that it is possible to automatically discover and learn vision-based reward functions from cross-embodiment demonstration videos that are robust to these differences. Specifically, we present a self-supervised method for Cross-embodiment Inverse Reinforcement Learning (XIRL) that leverages temporal cycle-consistency constraints to learn deep visual embeddings that capture task progression from offline videos of demonstrations across multiple expert agents, each performing the same task differently due to embodiment differences. Prior to our work, producing rewards from self-supervised embeddings typically required alignment with a reference trajectory, which may be difficult to acquire under stark embodiment differences. We show empirically that if the embeddings are aware of task progress, simply taking the negative distance between the current state and goal state in the learned embedding space is useful as a reward for training policies with reinforcement learning. We find our learned reward function not only works for embodiments seen during training, but also generalizes to entirely new embodiments. Additionally, when transferring real-world human demonstrations to a simulated robot, we find that XIRL is more sample efficient than current best methods. Qualitative results, code, and datasets are available at https://x-irl.github.io

**Keywords:** inverse reinforcement learning, imitation learning, self-supervised learning

## 1 Introduction

The ability to learn new tasks from third-person demonstrations holds the potential to enable robots to leverage the vast quantities of tutorial videos that can be gleaned from the world-wide web (e.g., YouTube videos). However, distilling diverse and unstructured videos into motor skills with vision-based policies can be daunting – as the videos themselves are often not only captured from different camera viewpoints in different environments, but also with different experts that may use different tools, objects, or strategies to perform the same task. Perhaps most critically, there often exists a clear *embodiment gap* between the human expert demonstrator, and the robot hardware that executes the learned policies. One approach to close this gap is to learn a mapping between the human and robot embodiment [1], which is a non-trivial intermediate problem in itself.

Despite considerable progress in learning policies with paired observations and actions (e.g., collected via teleoperation) [2, 3], much less work has been done in getting robots to learn policies from tasks defined only by third-person observations of demonstrations [4, 5, 6]. This is a surprisingly challenging problem, as different embodiments are likely to use unique strategies that suit them and allow them to make progress on a given task. For example, if asked to "place five pens into a cup", a human hand is likely to scoop up all pens before slipping them into the cup, whereas a two-fingered gripper might instead need to pick and place each pen individually. Both strategies complete the task, but generate different state-action trajectories. In this setting, it may be difficult to acquire labeled frame-to-frame correspondences between expert demonstration videos and learned embodiments [6], particularly across a multitude of embodiments or experts. The approach we investigate instead is whether we can successfully learn tasks through a learned notion of task progress that is invariant to the embodiment performing the task.

---

[*]Work done as an intern at Google.

5th Conference on Robot Learning (CoRL 2021), London, UK.

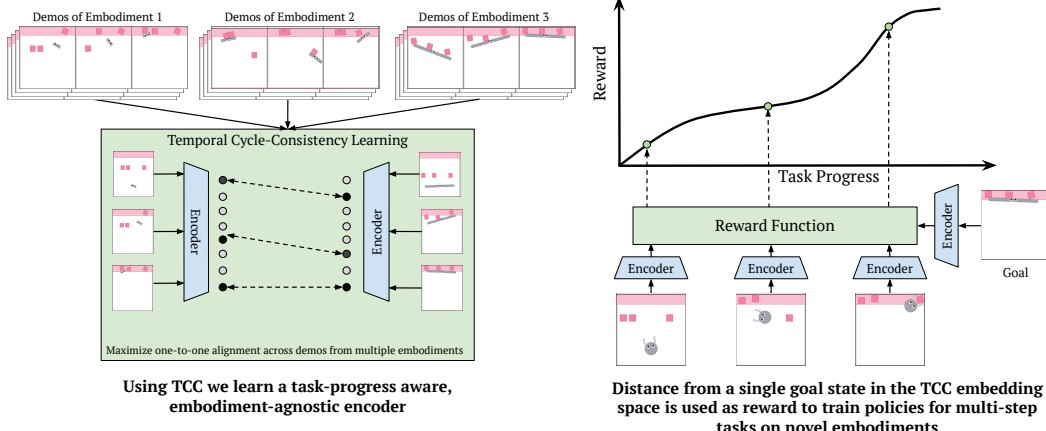

**Using TCC we learn a task-progress aware, embodiment-agnostic encoder**

**Distance from a single goal state in the TCC embedding space is used as reward to train policies for multi-step tasks on novel embodiments**

*Figure 1.* **Cross-embodiment Inverse Reinforcement Learning.** We learn embodiment-invariant visual representations from offline video demonstrations (stick agents on the left) using TCC [7], then use the trained encoder to generate embodiment-invariant visual reward functions that can be used to learn policies on new embodiments (gripper on the right) with reinforcement learning.

In this work, we propose to enable agents to imitate video demonstrations of experts – including ones with different embodiments – by using a task-specific, embodiment-invariant reward formulation trained via temporal cycle-consistency (TCC) [7]. We demonstrate how we can leverage an encoder trained with TCC to define dense rewards for downstream reinforcement learning (RL) policies via simple distances in the learned embedding space. Simulated experiments across four different embodiments show that these learned rewards are capable of generalizing to new embodiments, enabling unseen agents to learn the task via reinforcement, and surprisingly in some cases, exceeding the sample efficiency of the same agent learned with ground truth sparse rewards. We also demonstrate the effectiveness of our approach for learning robot policies using human demonstrations on the *State Pusher* environment from [6], where our reward is first learned on real-world human demonstrations, then used to teach a Sawyer arm how to perform the task in simulation.

Our contributions are as follows: **(i)** We introduce Cross-embodiment Inverse Reinforcement Learning (XIRL), an effective, label-free framework for tackling cross-embodiment visual imitation learning. Our core contribution is to use self-supervised learning on third-person demonstration videos to define dense reward functions amenable for downstream reinforcement learning policies, **(ii)** Along with XIRL, we release a cross-embodiment imitation learning benchmark, X-MAGICAL, which features multiple simulated agents with different embodiments performing the same manipulation task, including one thousand expert demonstrations for each agent, **(iii)** We show that XIRL significantly outperforms alternative methods on both the X-MAGICAL benchmark and the human-to-robot transfer benchmark from [6], and discuss our observations, which point to interesting areas for future research, **(iv)** Finally, we introduce a real-world dataset, X-REAL (Cross-embodiment Real demonstrations), of a manipulation task performed with nine different embodiments, which can be used to evaluate cross-embodiment reward learning methods[2].

## 2 Related Work

Traditional formulations of imitation learning [8, 1, 9] assume access to a corpus of expert demonstration data which includes both state and action trajectories of the expert policy. In the context of third-person imitation learning, including when learning from expert agents with different embodiments, obtaining access to ground-truth actions is difficult or impossible[3].

**Inferring expert actions.** To address this issue, several approaches either try to infer expert actions [10, 11, 12] – for example by training an inverse dynamics model on agent interaction data [10] – or employ forward prediction on the next state to imitate the expert without direct action supervision [13]. While these methods successfully address learning from observation-only demonstrations, they either do not support skill transfer to different policy embodiments at all, or they cannot take advantage of multiple embodiments in order to improve generalization to unseen policy configurations. We explicitly address these problems in this work.

**Imitation via learned reward functions.** In contrast to imitation via supervised methods, such as BCO [10], a recent body of work [4, 5, 14, 6, 15] has focused on learning reward functions from expert video data and then training RL policies to maximize this reward. In [4], the authors combine ImageNet pre-trained visual features with an L2-norm distance reward to match policy and expert observations

---

[2]For more details regarding X-REAL, please see Appendix C.

[3]For a more comprehensive related work, please see Appendix A.

in a latent feature space. In their follow-up work [5], the reward is computed in a viewpoint-invariant representation that is self-supervised on video data. While both these methods are compelling in their use of cheap unlabeled data to learn invariant rewards, the use of a time index as a heuristic for defining weak correspondence is a constraining limitation for tasks that need to be executed at different speeds, or are not strictly monotonic (e.g., have ambiguous sub-task ordering). In [14] a dense reward is learned via unsupervised learning on YouTube data and the authors make no assumption about time alignment. However, in their work, the expert and learned policy are executed in the same domain and embodiment, an assumption we relax in our work. Framed in a multi-task learning setting, [16] propose training policies with morphologically different embodiments first on a similar set of proxy tasks, in order to learn a latent space to map between domains, and then sharing skills on a held-out task from one policy to another. A time-index heuristic is used to define a metric reward when performing RL training of the new task. In our work, the learned embedding finds correspondences in a fully-unsupervised fashion, without the need for such strict time alignment. In [17], a small sub-set of states is human labeled for goal success and a convolutional network is then trained to detect successful task completion from image observations, where on-policy samples are used as negatives for the classifier. By contrast, our learned embedding encodes task progress in its latent representation without the use of expensive human labels.

**Imitation via domain adaptation.** An additional category of approaches to third-person imitation learning are those that perform domain adaptation of expert observations [18, 19, 20, 21]. For instance, in [18] a CycleGAN [22] architecture is used to perform pixel-level image-to-image translation between policy domains, which is then used to construct a reward function for a model-based RL algorithm. A similar model-free approach is proposed in [19]. In [20], a generative model is used to predict robot high-level sub-goals conditioned on a third-person demonstration video, and a lower-level control policy is trained in a task-agnostic manner. Similarly, [21] uses high level task conditioning from zero-shot transfer of expert demonstrations, but they use KL matching to perform both high and low-level imitation. In contrast to these methods, the unsupervised TCC alignment in this work avoids performing explicit domain adaptation or pixel-level image translation by instead learning a robust and invariant feature space in a fully offline fashion.

**Reinforcement learning with demonstrations.** Recent work in offline-reinforcement learning [6] explicitly tackles the problem of policy embodiment and domain shift. Their method, Reinforcement Learning from Videos (RLV), uses a labelled collection of expert-policy state pairs in conjunction with adversarial training to learn an inverse dynamics model jointly optimized with the policy. In contrast, we avoid the limitation of collecting human-labeled dense state correspondences by using a self-supervised algorithm (i.e., TCC [7]) which uses cycle-consistency to automatically learn the correspondence between states of two domains. We also show that this formulation improves generalization to unseen embodiments. Since the problem setup is similar to ours, we also compare to their method as a baseline.

## 3 Approach

Our overall XIRL framework (Figure 1) addresses the *cross-embodiment visual imitation* problem (Section 3.1). The framework consists of first using TCC to self-supervise embodiment-invariant visual representations (Section 3.2), then using a novel embodiment-invariant visual reward function (Section 3.3) to perform cross-embodiment visual imitation via reinforcement learning (Section 3.4).

### 3.1 Problem Formulation

Our objective is to extract an agent-invariant definition of a task, via a learned reward function, from a dataset of videos of separate agents performing the same task. In particular, we are interested in agents that may solve the task in entirely different ways due to differences in their end-effector, shapes, dynamics, etc., which we refer to as *embodiment* differences. For example, consider how differently a vacuum gripper and a parallel-jaw gripper will grasp an object as a result of their respective end-effectors. Such a setup is quite common in robotic imitation learning, where we might have access to observation data of humans demonstrating a task, but want to teach a robot to perform it. It is very likely that the way the human executes the task will diverge from how the robot would execute it.

We define a dataset of multiple agents performing the same task $T$ as $D = \bigcup_{i=1}^{n} D_i$, where $D_i$ is an agent-specific dataset containing observations of *only* agent $i$ performing task $T$. Each agent's dataset $D_i$ is a collection of videos defined as $D_i = \{v_i^1, v_i^2 ... v_i^K\}$, where $v_i^j$ represents the video of the $j^{th}$ demonstration of agent $i$ successfully performing the task $T$. We would like to highlight that $D$ only contains observation data, i.e., it does not store the actions taken by the respective agents. We use self-supervised representation learning techniques to learn task-specific representations from this dataset.

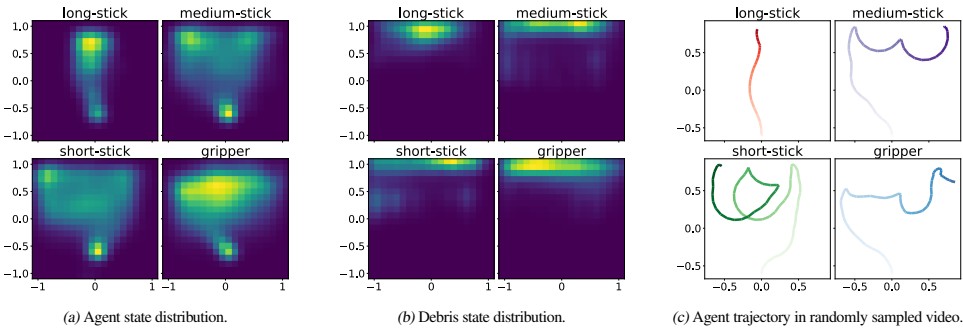

*(a) Agent state distribution.*     *(b) Debris state distribution.*     *(c) Agent trajectory in randomly sampled video.*

*Figure 2.* **Difference in state distributions** of the across different embodiments performing the same task in X-MAGICAL (Section 4.1).

## 3.2 Representation Learning

In this work, we use TCC [7] to learn task-specific representations in a self-supervised way. The method has been shown to learn useful representations on videos of the same action for temporally fine-grained downstream tasks. In their paper, the authors show that TCC representations can predict frame-level task progress, such as predicting how much water is in a cup during pouring, without requiring any human annotations. Task progress can provide dense signals for learning a new task and we would like to bake this property into our learned reward. Another advantage of TCC is that it does not require supervision for frame-level alignment (i.e., which frames in two videos correspond to each other). Such frame-to-frame correspondences are required by a prior method, RLV [6], to achieve successful reinforcement learning on the considered tasks, but we would like to avoid this type of manual supervision.

We train an image encoder $\phi$, that ingests an image $I$ and returns an embedding vector $\phi(I)$, using TCC. TCC assumes that there exist semantic temporal correspondences between two video sequences, even though they are not labeled and the actions may be executed at different speeds. Note that in our case, the assumption holds since all sequences in our dataset originate from the same task. Furthermore, even if two different agents accomplish the task very differently, there will be a common set of states or frames that will temporally correspond. When we apply the TCC loss, we compare frames of one agent performing the task with frames from another, and search for temporal similarities in how both execute the task. By performing multiple such comparisons and *cycling-back* (described in the next paragraph), TCC encodes task progress in its latent representation, a property that is useful for learning task-specific, yet embodiment-invariant reward functions. For completeness, we describe the training technique below.

We define a dataset $D_{\text{all}} = \{v_1, v_2 \ldots v_N\}$ that contains the videos of all agents executing the same task $T$. We are able to merge the datasets of different agents since TCC does not require embodiment labels (i.e., IDs) during training. We first sample a random batch of videos and embed all their frames using the aforementioned image encoder $\phi$. For each video $v_i$, this results in a sequence of embeddings $V_i = \{\phi(v_i^1), \phi(v_i^2) \ldots \phi(v_i^{L_i})\}$, where $L_i$ is the length of the $i^{th}$ video. From this mini-batch of sequences of video frame embeddings, we choose a pair of sequences $V_i$ and $V_j$ and compute their TCC loss. In particular, we randomly sample a frame embedding from sequence $V_i$ – say $V_i^t$ corresponding to the $t^{th}$ frame of video $v_i$ – and compute the soft nearest-neighbor of $V_i^t$ in sequence $V_j$ in the embedding space as follows:

$$\widetilde{V_{ij}^t} = \sum_{k}^{L_j} \alpha_k V_j^k, \quad \text{where} \quad \alpha_k = \frac{e^{-\|V_i^t - V_j^k\|^2}}{\sum_k^{L_j} e^{-\|V_i^t - V_j^k\|^2}}$$

We then *cycle-back* [7] to the first sequence $V_i$ by computing the soft-nearest neighbor of $\widetilde{V_{ij}^t}$ with all the frames in $V_i$. The probability of cycling-back to the $k^{th}$ frame in $V_i$ can be computed as:

$$\beta_{ijt}^k = \frac{e^{-\|\widetilde{V_{ij}^t} - V_i^k\|^2}}{\sum_k^{L_i} e^{-\|\widetilde{V_{ij}^t} - V_i^k\|^2}}$$

The expected frame index we cycle-back to is then $\mu_{ij}^t = \sum_k^{L_i} \beta_{ijt}^k k$. Since we know the index of the frame that started the cycle, in this case $t$, we can minimize the mean-squared (MSE) error loss between $t$ and the closest index retrieved via soft-nearest neighbor, i.e., $\widetilde{V_{ij}^t}$. The loss for a single frame is thus: $L_{ij}^t = (\mu_{ij}^t - t)^2$. Finally, we minimize the average loss $L$ over all frames in video $v_i$ with all other videos in the dataset $v_j$, defined as $L = \sum_{ijt} L_{ij}^t$.

*Table 1.* Statistics of Demos of the X-MAGICAL Embodiments

| Embodiment | Mean ± Std. Dev. Demo Length (no. of frames) |
|---|---|
| Long-stick | 48.0 ±28.5 |
| Medium-stick | 59.9 ±27.9 |
| Short-stick | 81.7 ±35.0 |
| Gripper | 110.8 ±39.8 |

### 3.3 Reward Function

Once the encoder $\phi$ has been trained on demonstrations of different agents performing the same task $T$, we want to use it to transfer information about the task from one agent to another. We do this by leveraging $\phi$ to generate rewards via distances to goal observations in the learned embedding space. Specifically, we define the goal embedding $g$ as the mean embedding of the last frame of all the demonstration videos in our offline dataset $D_{\text{all}}$. Concretely, $g = \sum_{i=1}^{N} \phi(v_i^{L_i})/N$, where $L_i$ is the length of video $v_i$. Our reward $r$ then is the scaled negative distance of the current state embedding to the goal embedding $g$ i.e., $r(s) = -1/\kappa \cdot \|\phi(s) - g\|_2^2$, where $\phi(s)$ is the state embedding at the current timestep and $\kappa$ is a scale parameter that ensures the distances are in a range amenable for training reinforcement learning algorithms [23]. We found it effective to set $\kappa$ to be the average distance of the first frame's embedding to the goal embedding for all the demonstrations in the dataset. Defining $r$ in this manner gives us several advantages: (a) it is dense, encoding both task completion and task progress, (b) it does not require any correspondence with a reference trajectory [5, 14] and (c) it sidesteps the need for a finite library of reference trajectories, unlike prior work [5, 16] that define time-indexed rewards relative to some reference trajectory. Thus, agents with trajectories of varying lengths (due to embodiment-specific constraints) can efficiently leverage this reward because the learned encoder can map different strategies to a common notion of task progress.

### 3.4 Reinforcement Learning

Using the pre-trained frozen encoder $\phi$, we define a Markov Decision Process (MDP) for any agent as the tuple $\langle \mathcal{S}, \mathcal{A}, P, r \rangle$ where $\mathcal{S}$ is the set of possible states, $\mathcal{A}$ is the set of possible actions, $P$ is the state transition probability matrix encoding the dynamics of the environment (including the agent) and $r$ is the *learned* reward function (defined in Section 3.3). Notice how we are able to use the same reward function $r$ for any agent – even ones that the encoder may not have seen during training. Furthermore, this reward function solely depends on the learned encoder. Hence, the task represented by the MDP now depends solely on how well the encoder has learned task-specific representations since we do not use the environment reward to either define the task or learn the policy. This is an important distinction because we are expecting the encoder to generalize to new states that an agent might encounter during training, as the expert demonstrations in our dataset only contain successful trajectories. It is also possible to augment sparse rewards (which only define task success or failure) with our learned dense reward while training policies.

## 4 Experimental Setup

### 4.1 X-MAGICAL Benchmark

We introduce a cross-embodiment imitation learning benchmark, X-MAGICAL, which is based on the imitation learning benchmark MAGICAL [24], implemented on top of the physics engine PyMunk [25]. In this work, we consider a simplified 2D equivalent of a common household robotic sweeping task, wherein an agent has to push three objects into a predefined zone in the environment (colored in pink). We choose this task specifically because its long-horizon nature highlights how different agent embodiments can generate entirely different trajectories. The reward in this environment is defined as the fraction of debris swept into the zone at the end of the episode.

**Multiple Embodiments in X-MAGICAL.** We create multiple *embodiments* by designing agents with different shapes and end-effectors that induce variations in how each agent solves a task. In Figure 1, we show three of these embodiments and some sample trajectories that solve the sweeping task. Please see Appendix B for a detailed description of the benchmark. Three agents are shaped like a *stick* and they differ only in length. We call them *short-stick*, *medium-stick* and *long-stick* based on the length of their body. The agent in the last row is called *gripper*: it is circular in shape and has two arms that can actuate. All agents are capable of two actions - a rotation around their axis and a translation in a forward/backward direction along this axis (similar to the agent in the MAGICAL benchmark). All agents have a two-dimensional action space and use force and torque control to change their position and orientation respectively. The

gripper agent has an additional degree of freedom for opening or closing its fingers. The default state of the gripper's fingers is open. For all agents, the state representation is a 16-dimensional vector with the following information: $(x, y)$ position of the agent, $(\cos\theta, \sin\theta)$ where $\theta$ is the agent's 2D orientation, and for each of the three debris: its $(x, y)$ position, its distance to the agent and its distance to the goal zone. We frame-stack [26] three consecutive state vectors to encode temporal and velocity information, resulting in a final state dimension of 48.

**Demonstrations and Different Embodiment Strategies in X-MAGICAL.** To learn task-specific representations for this task, we collect 1000 demonstrations per agent, where each demonstration consists in sweeping all three debris, initialized with random positions, into the target zone. This is the dataset $D_{\text{all}}$ (described in Section 3.1) containing observation-only agent-specific demonstrations. In Figure 2, we highlight the differences that exist between the trajectories taken by these agents. Figure 2a shows a heat map of the frequency of visits (*state visitation count*) of each agent at every 2D position in the grid, across all demonstrations in the dataset. We plot the 2D projection of the state visitation count onto the $XY$ plane with a bin width of 0.1. Yellow encodes higher state visitation whereas blue encodes lower state visitation. We observe that agent *long-stick* has less coverage of the environment as opposed to agent *gripper*, which has significantly more coverage. Similarly, we show the distribution of debris locations for all agents across all demonstrations in Figure 2b. In Figure 2c, we plot a randomly sampled trajectory from each agent's demonstration pool. We use transparency to encode the start (lighter) and end (darker) of the trajectory. It is clear from this figure that each agent solves the task in a different manner. Additionally, there is a significant difference in the time taken by each agent to execute this task, as shown in Table 1. Agent *long-stick* is able to finish the task the quickest because of its long shape that can sweep all the debris at once, while agents *short-stick* and *gripper* take longer because they have to frequently push or grasp one debris at a time. These differences are the types of challenges that a representation must overcome to successfully generalize across embodiments. As such, X-MAGICAL serves to create a highly simplified version of a real-world scenario where we might want to learn new tasks from an observation dataset of humans performing these tasks in highly diverse ways.

## 4.2 Baselines

Here, we describe the alternative reward functions we baseline our method against, color coded to match their appearance in Figs. 3, 4, and 5. *1) ImageNet*: We use an ImageNet pre-trained ResNet-18 encoder with no additional self-supervised training, i.e., we load the pre-trained weights, discard the classification head, and use the 512-dimensional embedding space from the previous layer. *2) Goal classifier*: We follow [27] and train a goal frame classifier on a binary classification task where the last frame of all the demonstrations is considered positive and all the others are considered negatives. We use the output probabilities of the classifier as the reward function. *3) LIFS*: We implement the method from [16] which learns a feature space that is invariant to different embodiments using a contrastive loss function paired with an autoencoding loss. *4) TCN*: single-view Time-Contrastive Network (TCN) [5] with positive and negative frame windows of 1 and 4 respectively. For more details regarding baseline implementations, see Appendix D.1.1.

## 4.3 Implementation Details

Our encoder for all experiments and methods is a ResNet-18 [28] initialized with ImageNet pre-trained weights. We replace the classification head with an embedding layer outputting a 32-dimensional vector. The encoder is trained on images of resolution $224 \times 224$ with ADAM [29] and a learning rate of $10^{-5}$. Note that our learned reward is agnostic to the RL algorithm used – in this work, we opt for Soft-Actor Critic (SAC) [30], which is a reinforcement learning algorithm that has been successfully used to train policies for continuous control tasks [31]. Once the TCC encoder is trained, we use it to embed the observation frames as the agent interacts with the environment.

## 5 Experiments

We execute a series of experiments to evaluate whether the learned reward functions are effective at visual imitation. Specifically, our experiments seek to answer the following questions: first (Section 5.1), in the ***same***-embodiment case, where the demonstration dataset $D$ contains the embodiment of the learning agent, does our method enable successful reinforcement learning for that agent? Next (Section 5.2), we investigate our primary interest, the ***cross***-embodiment case, where the demonstration dataset $D$ does **not** contain the embodiment of the learning agent. To additionally test our approach using real-world data (Section 5.3), we use the dataset from [6] to leverage real-world human demonstrations to learn policies in simulation. Note that each embodiment's performance is evaluated over 50 episodes and all figures plot the mean performance over 5 random seeds, with a standard deviation shading of $\pm 0.5$. Videos of our results are in the supplementary video.

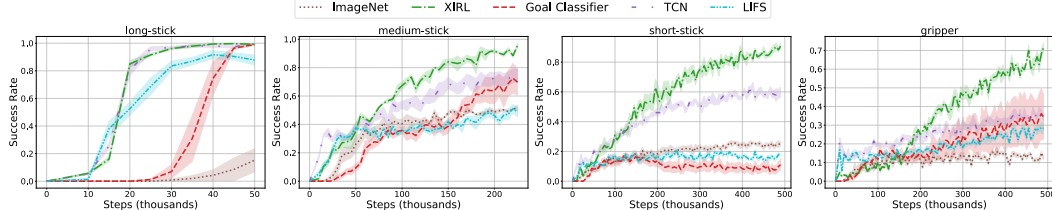

*Figure 3.* **Same-embodiment setting:** Comparison of XIRL with other baseline reward functions, using SAC [30] for RL policy learning on the X-MAGICAL *sweeping* task.

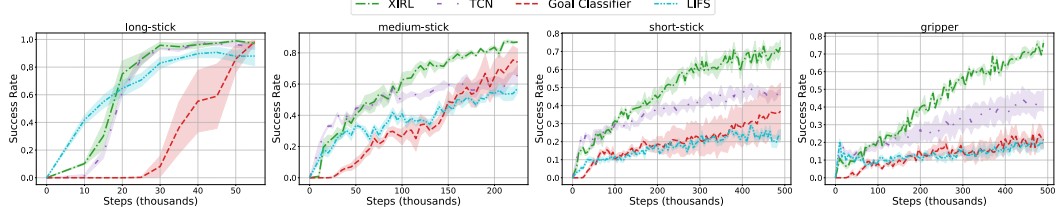

*Figure 4.* **Cross-embodiment setting:** XIRL performs favorably when compared with other baseline reward functions, trained on observation-only demonstrations from different embodiments. Each agent{*long-stick, medium-stick, short-stick, gripper*} is shown using demonstrations from the other 3 embodiments, with SAC [30] for RL policy learning on the X-MAGICAL *sweeping* task.

## 5.1 Results on Learning from Same-Embodiment Demonstrations

In this experiment, we want to validate whether our approach of using a reward function trained with TCC is good enough to train agents to perform the task defined in a dataset of expert demonstrations. Note that the learned reward function has to be robust enough such that it can provide a useful signal for new states an agent might encounter while learning a policy and interacting with the environment. In Figure 3, we compare our method XIRL with baselines described in Section 4.2. We find XIRL is more sample-efficient than the other learned reward baselines. We attribute this sample efficiency to the fact that the TCC embeddings encode task progress which helps the agent learn to reach for objects and goal zones while interacting with the environment, rather than exploring in a purely random manner. This experiment provides evidence that XIRL's reward function is suitable for downstream reinforcement learning.

## 5.2 Results on Learning from Cross-Embodiment Demonstrations

After verifying that XIRL works on the embodiments it was trained with, we move to the experiments that answer the **core question** in our work: *can XIRL generalize to unseen embodiments*? In this section we conduct experiments where the reward function is learned using embodiments that are different from the ones on which the policy is trained. As noted in Table 1, the timescales with which the agents execute the task can vary significantly. We conduct four experiments, each one corresponding to holding-out one agent from the expert demonstration set. In each such experiment, we train an encoder on demonstrations from the remaining three agents. We compare with reward functions learned using TCN, LIFS, and goal frame classifiers. While both TCC and TCN are contrastive losses, the former makes explicit comparisons across different embodiments, whereas the latter implicitly relies on an encoder shared across embodiments to learn the cross-embodiment representation. In Figure 4, we show that XIRL generalizes to new agents significantly better than the other learned reward baselines.

## 5.3 Results on Learning from Real-World Cross-Embodiment Demonstrations

In this experiment, we test how well we can learn rewards from more challenging real-world human demonstration videos. To do so, we use the dataset and *State Pusher* environment introduced in [6]. We train two XIRL encoders: XIRL (sim only) trained on 5 teleoperated simulated trajectories (i.e., no domain shift) and XIRL (real only) trained solely on the real-world human demonstrations without using any form of human labeling of paired frames. We compare our results to training the policy on the sparse reward from the environment and the RLV method presented in [6]. As demonstrated in Figure 5, we find our approach can improve the sample-efficiency of learning in the environment, compared to using RLV or solely the environment reward.

## 5.4 Qualitative comparison between learned reward functions vs. handcrafted rewards

In Figures 6 and 7, we visualize and compare the XIRL reward function with the environment's reward (ground truth) for the *Sweeping* task from X-MAGICAL (see Sec. 4.1 for details) and the *State Pusher* and *Drawer Opening* tasks from [6][4]. We find that the learned reward is highly correlated with the ground truth

---

[4]For additional qualitative visualizations on a real-world dataset with multiple embodiments, see Appendix C.

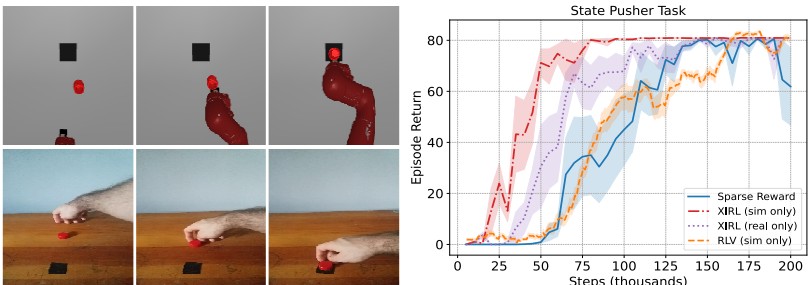

Figure 5. **Real-world-demo cross-embodiment setting:** Comparison of XIRL with baselines using the simulated *State Pusher* environment from [6]. XIRL (real only) leverages real-world demonstration videos of humans (left, row 2) to teach a robot arm in sim (left, row 1), but unlike [6], we do not use human-labeled data of paired frame correspondences. RLV* denotes results taken verbatim from [6] which uses a different implementation of SAC for RL policy learning.

reward from the environment for both successful demos (first column in Figures 6 and 7) and unsuccessful demos (second column in Figures 6 and 7). It is especially encouraging to see that in a sparser reward environment like the *Drawer Opening* task, XIRL provides a dense signal that should allow the agent to learn the task more efficiently. Additionally, we can see that in the example of the failed collision trajectory (i.e., second row third column in Fig. 7), where the arm collides with the drawer rather than opening it, XIRL is able to provide it with a partial reward (i.e., for correctly moving towards the drawer) as opposed to the environment reward which remains zero.

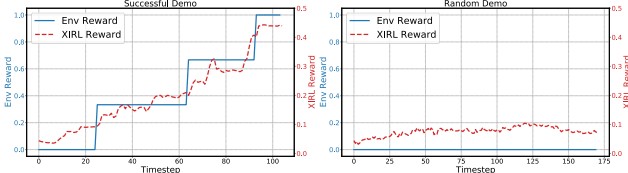

Figure 6. **X-MAGICAL demo cross-embodiment setting:** Visualizing our learned reward function XIRL vs. the environment's sparse reward on a successful and unsuccessful demonstration, for the *short-stick* agent on the X-MAGICAL *sweeping* task. The learned reward was trained on the three other agents.

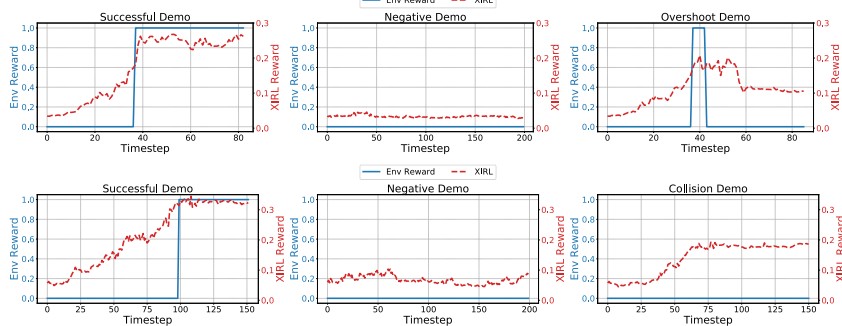

Figure 7. **Real-world-demo cross-embodiment setting:** Visualizing our learned XIRL reward function vs. the environment's sparse reward on the *State Pusher* (top) and *Drawer Opening* (bottom) tasks from [6].

## 6  Conclusion

This paper presents XIRL, a framework for learning vision-based reward functions from videos of expert demonstrators exhibiting different embodiments. XIRL uses TCC to self-supervise a deep visual encoder from videos, and uses this encoder to generate rewards via simple distances to goal observations in the embedding space. XIRL enables unseen agents with new embodiments to learn the demonstrated tasks via IRL. Reward functions from XIRL are fully self-supervised from videos, and we can successfully learn tasks without requiring manually paired video frames [6] between the demonstrator and learner. In this sense, our method presents favorable *scalability* to an arbitrary number of embodiments or experts with varying skill levels. Experiments show that policies learned via XIRL are more sample efficient than multiple baseline alternatives, including TCN [5], LIFS [16], and RLV [6]. While our experiments demonstrate promising results for learning policies in simulated environments using rewards learned from both simulated and real-world videos, we have yet to show policy learning on a real robot, which we look forward to trying post-COVID.

## Acknowledgments

We would like to thank Alex Nichol, Nick Hynes, Sean Kirmani, Brent Yi and Jimmy Wu for fruitful technical discussions, Sam Toyer for invaluable help with setting up the simulated benchmark, and Karl Schmeckpeper for discussions and help related to RLV.

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
