# OpenReview forum: "XIRL: Cross-embodiment Inverse Reinforcement Learning"
_robot-learning.org/CoRL/2021/Conference — CoRL2021 Oral_

### Official Review · Reviewer_tUkm · 2021-07-12

**Originality:** Fair
**Technical Quality:** Good
**Clarity Of Presentation:** Very Good
**Impact:** 2

**Recommendation:**

Weak Accept: I recommend accepting the paper, but will not argue for my recommendation if the majority of other reviewers have a different opinion.

**Summary:**

Paper proposes to use TCC (temporal cycle-consistency learning) for learning representations in cross-embodiment visual imitation learning tasks. The authors empirically demonstrate that the learned representations can deliver a simple reward function (the distance between the current state and goal representations) that is sufficient for RL. A cross-embodiment benchmark (X-MAGICAL) is also presented and the proposed approach attains superior performances on both the simulation and real-world data.

**Issues:**

I may consider further increasing my score if the following points can be addressed:

1. Citations to the aforementioned papers.

2. Extra results as requested above.

**Reviewer Expertise:**

Good: General knowledge of the area

**Strengths And Weaknesses:**

[Strengths]

[+] The paper is overall clear and well-written. I like how the authors approach their main results in an illustrative fashion and I believe this paper can also be easy to follow for readers without dense imitation learning background.

[+] The proposed method is well-motivated. I agree that cross-embodiment imitation learning can effectively be a representation learning problem and TCC seems to be a reasonable inductive bias to produce the needed representations. I'm a little bit surprised that the resulting representations can even provide a straightforward reward solution.

[+] I particularly like the real-world experiments and the results are truly impressive.

[Weaknesses]

I don't have major concerns at this point but I do encourage the authors to address the following issues in a rebuttal:

[-] It is still unclear how the agent can benefit from cross-embodiment pretraining. Indeed I think the crucial question is: does the agent learn from other embodiments or just learn from visual pretraining. The following experiments may help clarify this question:

- Add a baseline that just leverages general-purpose self-supervised visual pretraining, e.g. MoCo, SimCLR.
- Try to learn from other embodiments only in the pretraining stage.

[-] Some papers on imitation learning from observations (somehow shares the goal of cross-embodiment imitation learning) should be cited:

[1] Imitation Learning from Observations by Minimizing Inverse Dynamics Disagreement

[2] Learning and Inferring Movement with Deep Generative Model

[3] Third-Person Imitation Learning

[4] Provably Efficient Imitation Learning from Observation Alone

**Summary Of Recommendation:**

Given the overall technical quality, I recommend acceptance but I'm open to other assessment if there is something I missed.

---

> ### Author Response · Authors · 2021-08-29
> **Authors' Response for Reviewer tUkm**
>
> Thank you for taking the time to review our paper and for providing helpful comments! We are happy to hear that you found the paper well motivated and enjoyable to read. We are also pleased to hear that you appreciated the real world experiments. You brought up some helpful suggestions that we discuss below. Please let us know if you have any remaining questions or concerns.
>
> Comment: **It is still unclear how the agent can benefit from cross-embodiment pretraining. Indeed I think the crucial question is: does the agent learn from other embodiments or just learn from visual pretraining. The following experiments may help clarify this question:**
> 1. **Try to learn from other embodiments only in the pretraining stage.**
>
>     * Response: We agree with the reviewer about their suggestion that we must “try to learn from other embodiments only in the pretraining stage.” This is the objective of our cross-embodiment experiments (Section 5.2 and Figure 4). In the pretraining stage, we only train on other embodiments (holding out the test embodiment) and then use the learned encoder on the test embodiment for downstream RL. For example, in Figure 4a, longstick means the test embodiment was the longstick embodiment and pretraining was done on gripper, shortstick and mediumstick.
>
> 2. **Add a baseline that just leverages general-purpose self-supervised visual pretraining, e.g. MoCo, SimCLR.**
>
>     * Response: Thanks for suggesting that we add a baseline that leverages established self-supervised visual pretraining techniques. We are happy to report that we’ve added 2 new SimCLR baselines in the form of: i) a Resnet18 baseline trained with SimCLR on the x-MAGICAL dataset (green curve) and ii) a Resnet18 baseline that has been pretrained on ImageNet with SimCLR with no further pre-training on x-MAGICAL (orange curve). **Those results are now in Appendix E.2**. In the latter baseline, each frame in the dataset is considered an individual image on which we apply data augmentations. Additionally, every other sampled image in the batch is considered a negative. Here we discuss the main findings of the experiments, please see the appendix for the plots.We find that the SimCLR objective performs poorly when trained solely on the x-MAGICAL dataset while the SimCLR pertained on ImageNet (without finetuning on x-MAGICAL) does much better on the longstick embodiment (easier). For the mediumstick embodiment which is harder to solve, both perform poorly as compared to XIRL (shown in blue) which performs significantly better. Overall, both visual pretraining on cross-embodiment demonstrations and the inductive biases offered by the TCC loss are required to get good performance on downstream RL tasks.
>
> Comment: **Some papers on imitation learning from observations (somehow shares the goal of cross-embodiment imitation learning) should be cited [...].**
>
> * Response: Thanks for bringing up these papers. We’ve added them in the updated revision (see the expanded related works in Appendix A). Note that [3] had already been previously cited.
>
> Thanks again for your time reviewing our paper and for the helpful suggestions for improvement. We hope we’ve clarified your questions and concerns and that we’ve convinced you to increase your score!
>
> [3] Third-Person Imitation Learning

---

> ### Author Response · Authors · 2021-09-07
> **Post-rebuttal update from reviewer tUkm**
>
> We were wondering if reviewer tUkm has had a chance to look at the rebuttal updates and if they have helped clarify concerns and strengthen the paper.
>
> Best,
> Authors

---

### Official Review · Reviewer_dDLT · 2021-07-21

**Originality:** Very Good
**Technical Quality:** Very Good
**Clarity Of Presentation:** Very Good
**Impact:** 4

**Recommendation:**

Strong Accept: I recommend accepting the paper and will argue for my recommendation even if other reviewers hold a different opinion.

**Summary:**

This paper presents a method of learning cross domain representations.  These representations can be used as a reward signal to speed the training of RL policies.

**Issues:**

The paired data in RLV is in the expert domain, but is off policy.

What is the difference between Env Reward in table 2 and Env Reward Only in Figure 5?

I don't see the training curves for the Drawer Opening task, despite Figure 7 including it.

The X-Real tasks would be better in the main paper, rather than the supplemental.

**Reviewer Expertise:**

Excellent: Expert knowledge on the topic of the paper

**Strengths And Weaknesses:**

The paper is mostly clearly written and presents an interesting new method.

The performance using human data is impressive, both on the X-Real dataset and on the RLV tasks.

**Summary Of Recommendation:**

The paper presents a novel method for learning cross domain embeddings and reward signals and provides extensive experimentation to show that the proposed method is effective in a variety of environments.

---

> ### Author Response · Authors · 2021-08-29
> **Authors' Response for Reviewer dDLT**
>
> Thank you for taking the time to review our paper and for providing helpful comments. Here are a few responses to some of the comments you made. Please let us know if you have any remaining questions or concerns.
>
> Comment: **What is the difference between Env Reward in table 2 and Env Reward Only in Figure 5?**
>
> * Response: In Figure 5, note that both the RLV baseline and our XIRL method use the sparse environment reward plus the learned reward to train the policy. The “env reward only” baseline is a policy trained only with the sparse environment reward (i.e., with no human data). Note that in the x-MAGICAL experiments, “env reward” is the reward given by the environment and detailed/visualized in Appendix B.2.
>
> Comment: **I don't see the training curves for the Drawer Opening task, despite Figure 7 including it.**
>
> * Response: Yes, please note that the Drawer Opening task from RLV provides only visual observations, with no state observations. For the RL experiments in our paper we used (i) visual observations for the rewards and (ii) state observations into the (SAC-trained) policy, including the Puck Pushing task as was done in RLV. Figure 7 (bottom) included a visualization of the learned reward curve for Drawer Opening to show how well the XIRL encoder reward correlates with the environment reward. It also exhibits some desirable properties for downstream RL (e.g. smoothness). We are currently working on pixel-based policy experiments and hope to include them in the camera-ready.
>
> Comment: **The X-Real tasks would be better in the main paper, rather than the supplemental.**
>
> * Response: While there currently isn’t enough space to move it without exceeding the 8-page limit, we’ll try to squeeze/reshuffle things around for the camera ready. In the meantime, we’ve made a more prominent reference to X-REAL in the contribution section of the updated revision.
>
> Thanks again for your time reviewing our paper and for the helpful suggestions for improvement. We hope we’ve clarified your questions and concerns.

---

### Official Review · Reviewer_Fbyh · 2021-07-26

**Originality:** Good
**Technical Quality:** Excellent
**Clarity Of Presentation:** Excellent
**Impact:** 4

**Recommendation:**

Strong Accept: I recommend accepting the paper and will argue for my recommendation even if other reviewers hold a different opinion.

**Summary:**

This work introduces the use of Temporal Cycle-Consistency (TCC) for imitation learning. By training an encoder via TCC to embed similar states across different embodiments into similar latent variables, the authors can use proximity in latent space to a goal state as reward for imitation. The authors show this to great success in an artificial sim-to-sim 2D task and to decent success in a real-to-sim task.

To me, this paper is pretty water-tight. The method isn't groundbreaking because it's mostly using TCC, but the idea of using that as reward is novel enough to warrant publication. The experiments are all solid, the baselines are solid, and I'm honestly failing to see any major weaknesses. So I'm recommending acceptance.

**Issues:**

Please see weakness section

**Reviewer Expertise:**

Very good: Comprehensive knowledge of the area

**Strengths And Weaknesses:**

### Strengths:
- Great experiments. 2D sim2sim environment is a great illustration and quick turnaround cross-embodiment playground.
- Writing is great and clear. The background section is just right for my personal taste in terms of explaining the basics but not overexplaining anything.
- In the appendix there is a bonus dataset that should be put into the main body of the paper, X-REAL, that looks fun.
- Very specific about the implementation and training details (in main paper and appendix), great work!
- Using 5 seeds for all RL experiments, good job!
- The smooth (learned) rewards in Fig. 6 and 7 are excellent.

### Weaknesses:

(None of these are major, just little things that I think might improve the paper)

1. Maybe also include or at least mention VIRL (https://arxiv.org/abs/1901.07186), since it's similar to TCN and needs fewer expert samples. Compared to your method, it takes an embedding of the entire sequence so far and compares it to the expert.
2. Fig.1 left side, bottom - what's that extra encoder at the bottom, and what's its output? How's it trained? Isn't that encoder part of the TCC training process? If that's the case, just remove it.
3. In line 123 you refer to Fig.2c but the task wasn't really introduced yet. So to the average reader, up until then, Fig.2c is just squiggly lines.
4. line 169 - you're using the mean embedding of all agents' last state as goal? Couldn't that potentially lead to an unreachable goal? If I look for example at the t-SNE embedding of the real videos, some of the final states are very far apart. Maybe as an analogy, when generating new images with a VAE by interpolating between two latents, when moving linearly through the state space, the results are usually unrealistic-looking, but when moving across a hypersphere between both samples, the results look much better (see here for example https://arxiv.org/abs/1609.04468). Maybe it'd be better to take the middle point on a hypersphere instead of the euclidean mean?

**Summary Of Recommendation:**

As mentioned above, the paper is very solid.

---

> ### Author Response · Authors · 2021-08-29
> **Authors' Response for Reviewer Fbyh**
>
> Thank you for taking the time to review our paper and for providing helpful comments! We are happy to hear that you found the paper solid overall, enjoyable to read, and that its technical quality was excellent. We are also pleased to hear that you appreciated the value of our environments: (a) x-MAGICAL for providing a “*quick turnaround cross-embodiment playground*” (b) and x-REAL for a “*fun*” real-world dataset. You brought up some helpful suggestions for improvements that we discuss below. Please let us know if you have any remaining questions or concerns.
>
> Comment: **In the appendix there is a bonus dataset that should be put into the main body of the paper, X-REAL, that looks fun.**
>
> * Response: Thank you for the kind words. While there currently isn’t enough space to move it without exceeding the 8-page limit, we’ll try to squeeze/reshuffle things around for the camera ready. In the meantime, we’ve made a more prominent reference to X-REAL in the contribution section of the updated revision.
>
> Comment: **Maybe also include or at least mention VIRL (https://arxiv.org/abs/1901.07186), since it's similar to TCN and needs fewer expert samples. Compared to your method, it takes an embedding of the entire sequence so far and compares it to the expert.**
>
> * Response: Thanks for bringing up VIRL, we agree that it is very relevant. We’ve added it to the related works of the updated revision.
>
> Comment: **Fig.1 left side, bottom - what's that extra encoder at the bottom, and what's its output?**
>
> * Response: It’s the same encoder, we only meant to illustrate that after pretraining, we produce/output a pretrained TCC encoder. We can see how this may cause confusion and have removed it in the updated revision.
>
> Comment: **In line 123 you refer to Fig.2c but the task wasn't really introduced yet. So to the average reader, up until then, Fig.2c is just squiggly lines.**
>
> * Response: We completely agree, we’ve eliminated the comment in the updated draft.
>
> Comment: **You're using the mean embedding of all agents' last state as a goal? Couldn't that potentially lead to an unreachable goal? If I look for example at the t-SNE embedding of the real videos, some of the final states are very far apart. [...] Maybe it'd be better to take the middle point on a hypersphere instead of the euclidean mean?**
>
> * Response: Thanks for this great suggestion! We do agree that it’s theoretically possible for the mean embedding to produce an unreachable state. However, we note that empirically we found that it worked quite effectively for the experiments demonstrated in the paper. We’ll look into using this version as well in future revisions of the paper.
>
> Thanks again for your time reviewing our paper and for the helpful suggestions for improvement. We hope we’ve clarified your questions and concerns.

---

### Meta-Review · Area_Chair_HnWL · 2021-08-13

**Recommendation:** Accept (Oral)
**Confidence:** 4

**Metareview:**

This paper proposed to learn cross domain representations as a reward signal to achieve RL policies, and present a self-supervised method to learn deep visual embeddings considering embodiment differences.

I agree with reviewers that this paper is well organized and the idea is quite novel, the extensive experiments demonstrated the proposed method is effective in a variety of environments.

I believe this is a good work, I agree with reviewers on some minor points they mentioned, and suggest authors carefully address these comments to further improve this paper.

More added:
I would like to thank authors for your carefully and detailed responses to comments raised by reviewers, I really believe the revised version will be more better. Congratulations!

---

> ### Author Response · Authors · 2021-08-29
> **Authors' Response to the Meta Review**
>
> Thank you for your time and kind words, Area Chair. We have made updates to the paper and supplementary material (mentioned in reviewer responses), and have also replied to each reviewer’s comments. We are also pleased to report that we’ve improved the success curves for XIRL in Section 5.2, Figure 4 and Section 5.3, Figure 5. Here is a summary of our primary responses to reviewers’ questions. More details can be found in the individual responses to each reviewer.
>
> 1. **New Experiments: Added 2 SimCLR baselines for both same and cross embodiment settings**
>
> Reviewer *tUkm* raised the point that it was still unclear whether the benefit of our method is coming from the visual pre-training aspect or from the cross-embodiment demonstrations.
>
> They suggested we try “to learn from other embodiments only in the pre-training stage”. We note that Section 5.2 (and Figure 4) of the paper studies this exact setting, denoted “cross-embodiment” in the paper, where the agent embodiment on which RL is performed is **held out** of the pre-training dataset and thus tests XIRL’s ability to generalize to the unseen embodiment.
>
> They also suggested trying out baselines that “just leverage[s] general-purpose self-supervised visual pre-training, e.g. MoCo, SimCLR”. We are happy to report that we’ve implemented the suggestion in the form of **two new baselines** to both same and cross embodiment settings. They are: i) a Resnet18 baseline trained with SimCLR on the x-MAGICAL dataset and ii) a Resnet18 baseline that has been pre-trained on ImageNet with SimCLR with no further pre-training on x-MAGICAL. **Those results are now in Appendix E.2**. They show that XIRL significantly outperforms both baselines on the two agent embodiments we tested.
>
> 2. **Moving X-REAL to the main body of the paper**
>
> Reviewers *dDLT* and *Fbyh* appreciated the real-world X-REAL dataset and experiment and suggested we move it to the main body of the paper. While there currently isn’t enough space to do so without exceeding the 8-page limit, we’ll try to squeeze/reshuffle things around for the camera-ready. In the meantime, we’ve made a more prominent reference to X-REAL in the contribution section of the updated revision.
>
> 3. **Citing more references**
>
> Reviewers *dDLT* and *tUkm* suggested we cite a few more relevant papers. We have added these papers to the Expanded Related Works section in Appendix A of the updated revision.

---

### Decision · Program_Chairs · 2021-09-13

**Decision:**

Accept (Oral)

**Comment:**

This paper proposed to learn cross domain representations as a reward signal to achieve RL policies, and present a self-supervised method to learn deep visual embeddings considering embodiment differences.

I agree with reviewers that this paper is well organized and the idea is quite novel, the extensive experiments demonstrated the proposed method is effective in a variety of environments.

I believe this is a good work, I agree with reviewers on some minor points they mentioned, and suggest authors carefully address these comments to further improve this paper.

More added:
I would like to thank authors for your carefully and detailed responses to comments raised by reviewers, I really believe the revised version will be more better. Congratulations!